# Molecular Doping for Hole Transporting Materials in Hybrid Perovskite Solar Cells

**Vanira Trifiletti** [1,2,3,*], **Thibault Degousée** [1], **Norberto Manfredi** [2], **Oliver Fenwick** [1], **Silvia Colella** [3,4] **and Aurora Rizzo** [4]

1  School of Engineering and Materials Science (SEMS), Queen Mary University of London, Mile End Road, London E1 4NS, UK; t.degousee@qmul.ac.uk (T.D.); o.fenwick@qmul.ac.uk (O.F.)
2  Department of Materials Science and Milan-Bicocca Solar Energy Research Center–MIB-Solar University of Milan-Bicocca, Via Cozzi 55, 20125 Milano, Italy; norberto.manfredi@unimib.it
3  Department of Mathematics and Physics "E. De Giorgi" University of Salento, 73100 Lecce, Italy; silvia.colella@unisalento.it
4  Istituto di Nanotecnologia CNR-Nanotec, Polo di Nanotecnologia c/o Campus Ecotekne via Monteroni, 73100 Lecce, Italy; aurora.rizzo@nano.cnr.it
*  Correspondence: v.trifiletti@qmul.ac.uk; Tel.: +44-(0)20-7882-5306

**Abstract:** Hybrid lead halide perovskites have been revolutionary in the photovoltaic research field, reaching efficiencies comparable with the most established photovoltaic technologies, although they have not yet reached their competitors' stability. The search for a stable configuration requires the engineering of the charge extraction layers; in this work, molecular doping is used as an efficient method for small molecules and polymers employed as hole transport materials in a planar heterojunction configuration on compact-$TiO_2$. We proved the viability of this approach, obtaining significantly increased performances and reduced hysteresis on compact titania-based devices. We investigated the photovoltaic performance correlated to the hole transport material structure. We have demonstrated that the molecular doping mechanism is more reliable than oxidative doping and have verified that molecular doping in polymeric hole transport materials leads to highly efficient perovskite solar cells, with long-term stability.

**Keywords:** molecular doping; perovskite solar cell; stability; hysteresis; F4-TCNQ

## 1. Introduction

Hybrid lead halide perovskites are revolutionary materials for a wide range of optoelectronic and electronic applications [1–3]. In the photovoltaic research field, they have shown efficiencies comparable with the most established photovoltaic technologies, overcoming the thin-film technologies, although perovskite-based solar cells have not yet reached their competitors' stability [1,4–6]. A step towards achieving devices with long-term stability has recently been achieved with mixed-cation lead mixed-halide perovskites, such as $(HC(NH_2)_2)_{0.83}Cs_{0.17}Pb(I_{0.6}Br_{0.4})_3$ [7,8]. Therefore, the search for the most stable configuration is, nowadays, dominated by the search for charge extractor materials that can stabilise the solar cell performance [4,9]. In conventional device architecture (substrate, cathode, electron transporting material (ETM), hybrid perovskite, hole transporting material (HTM), anode), the most commonly employed HTM, due to its significant solubility and excellent hole mobility, is Spiro-OMeTAD ((2,2′,7,7′-Tetrakis N, N-di-p-methoxyphenylamine)-9,9′-spirobifluorene) [10,11]. Spiro-OMeTAD has given excellent results—the doping occurs by the oxidation of the complex that it forms with the additive lithium bis(trifluoromethanesulfonyl)imide (LiTFSI) in an open system condition. However, it is hard to collect consistent results, because the amount of oxidised

Spiro-OMeTAD depends on external factors (e.g., oxygen in the active layer, light intensity, humidity, and room temperature). The fluctuation of the oxidised Spiro-OMeTAD concentration affects device stability and reproducibility [12], and the Li+ cations can easily migrate across the perovskite film and reach the ETM interface, with a massive impact on the hysteresis [13].

In order to address this issue, other organic materials have been successfully employed as HTMs in perovskite solar cells, such as poly(3-hexylthiophene-2,5-diyl) (P3HT) [14] which has a higher hole mobility (up to 0.1 cm$^2$ V$^{-1}$ s$^{-1}$ in LiTFSI-doped P3HT compared to $10^{-4}$ cm$^2$ V$^{-1}$ s$^{-1}$ in LiTFSI-doped Spiro-OMeTAD) [15]. Lately, polymeric HTMs are attracting attention, because the film structure is not affected by the working operation condition, and they can form an adequate barrier to prevent gold migration from the back contact [16]. With the aim of decreasing production costs, interest in the study of P3HT as a HTM in perovskite solar cells has increased; it is a well-known material, and it is easy to coat [14,17–20]. Undoped P3HT has proven to be extremely effective; when it was tested on perovskite films with very low defect densities and an engineered interface, it achieved high solar conversion efficiencies [14,21]. Still, long-term stability remains an issue. The most remarkable result to date is that after 1008 h under 85% relative humidity at room temperature P3HT retained 80% of its first efficiency [14]. A way to increase stability is to exploit the molecular doping methodology. Doping technology is extensively used in photovoltaic and electroluminescent devices, based on inorganic, organic and hybrid materials, to enhance the conductivity of the transporting layers [22]. Additionally, doping has proven to be an effective way to increase stability in organic photovoltaic devices, and a large variety of molecular species can serve as dopants in organic semiconductors [23–25].

In this study, 2,3,5,6-Tetrafluoro-7,7,8,8-tetracyanoquinodimethane (F4-TCNQ) was employed [26–29] as an additive in a small molecule HTM (Spiro-OMeTAD) and polymer HTM (P3HT). F4-TCNQ is a strong electron acceptor and is widely used as a hole selective layer or as an additive in p-type doping [26,30–36]. F4-TCNQ is commonly used to minimise the hole injection barrier, and it has been proven that F4-TCNQ doping generates a polaron charge-transfer complex that increases HTM conductivity and decreases the charge recombination [37]. Huang et al. [38] reported using F4-TCNQ:Spiro-OMeTAD as a HTM layer which improved the air stability of perovskite solar cells. Their device reached 10.6% photon conversion efficiency, PCE, comparable to LiTFSI doping (PCE = 12.7%). However, the F4-TCNQ-based device retained 85% of this PCE after 72 h and 60% after one week of the first performance, compared with the LiTFSI-based device that retained 20% after 72 h and 10% after one week of the initial performance. They attributed the increased stability to the substitution of hygroscopic LiTFSI with the non-hygroscopic F4-TCNQ. Encouraged by the results mentioned above, we tested F4-TCNQ doping, in various concentrations, in the planar heterojunction configuration on compact-TiO$_2$ (c-TiO$_2$:PHJ, Figure 1 in the relief); in this device architecture, the ETM is only composed of a layer of about 80 nm of titania, and no scaffold is used [39]. In order to obtain reliable CH$_3$NH$_3$PbI$_3$ layers, these layers were grown following the procedure published by the authors elsewhere [40]. As demonstrated by the high efficiencies that the fine engineering of the absorber layer and its interface with the charge extraction layers can provide [14,41,42], we employed a compact perovskite film with reduced surface defects and large grain size, in order to realise devices with a photovoltaic performance that is independent of the voltage scan direction. A scanning electron microscopy (SEM) image of the employed perovskite film is shown in Figure 1b.

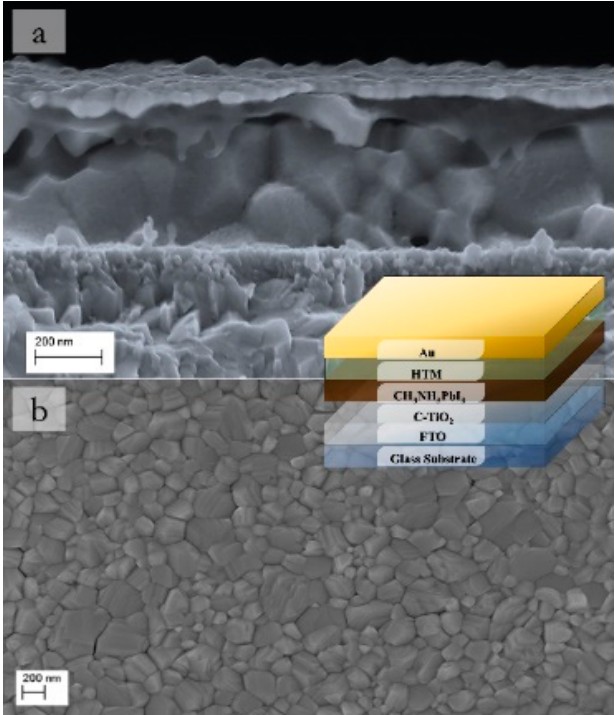

**Figure 1.** Scanning electron microscopy (SEM) images of (**a**) the device cross-section and (**b**) the employed $CH_3NH_3PbI_3$ perovskite film. Representation of the device structures in the relief.

## 2. Materials and Methods

Hybrid perovskite solar cells were prepared in a planar heterojunction configuration on compact-$TiO_2$ (c-$TiO_2$:PHJ) (relief in Figure 1), and LiTFSI-doped Spiro-OMeTAD was employed as a control device [40]. Fluorine tin oxide (FTO)/glass were employed for the substrate and partially etched. Substrate cleaning was performed using pure water, acetone and isopropanol in an ultrasonic bath. Next they were immersed in a TL1 washing solution, which was heated to 80 °C (10 min). Then they were rinsed using pure water. The $TiO_2$ was synthesised by twice spin coating 0.15 M of titanium diisopropoxidebis (acetylacetonate) solution in 1-butanol and once spin coating 0.3 M of titanium diisopropoxidebis (acetylacetonate) solution in 1-butanol. After each deposition, the sample was heated on a hot plate at 125 °C for 5 min and then heated for 2 h in an oven at 520 °C [40].

$CH_3NH_3I$ was synthesised according to the procedure reported in the literature [40]. The 2 M $CH_3NH_3PbI_3$ solution was prepared according to the procedure published by the authors elsewhere [40]: (i) the $PbI_2$ pellets were solved in a mixture of $\gamma$-butyrolactone and dimethyl sulfoxide for 1 h at 90 °C, and the $CH_3NH_3I$ powder was added, (ii) the mixture of $PbI_2$ pellets and $CH_3NH_3I$ powder was stirred for 30 min at 70 °C and kept overnight at 60 °C. The day after, 200 μL was dropped and spin-coated on the $TiO_2$/FTO at 1000 and 4000 rpm, respectively, for 20 and 60 s. During the final spin-coating step, 200 μL of dichloromethane was dropped. The samples were then annealed for 75 min at 100 °C on a hot plate and subjected to $9.0 \times 10^{-7}$ mBar for 4 h [40].

The Spiro-OMeTAD in chlorobenzene concentration was 63 mg mL$^{-1}$, and it was doped by tert-butylpyridine (27.8 μL) and LiTFSI in acetonitrile solution (30.7 μL). The spin rate was 2500 rpm for 45 s, in order to produce a layer of 90 nm. Spiro-OMeTAD doped with F4-TCNQ (Sigma-Aldrich) was prepared in 0.1, 0.5, 1 and 5 wt %. The doped solutions were spun at a rate of 2500 rpm for 45 s. We, therefore, extended this approach to P3HT (Regioregularity ≥ 98%, Sigma-Aldrich) as proof of concept, in which we used a 15 mg mL$^{-1}$ solution in chlorobenzene that was stirred at 80 °C for 10 min. Once cooled to room temperature, 100 μL of the filtered solution was spin-coated onto the perovskite film at 600 and 2000 RPM for 12 and 40 s, respectively [17]. The coated films were then annealed on

a hot plate set at 100 °C for 10 min to increase the order in the P3HT film structure [43]. The P3HT film thickness was about 60 nm. Various amounts of F4-TCNQ were added to the P3HT solution to change the doping concentration. The coating of the perovskite films was unchanged for both types of HTM-doped layers. Finally, 80 nm Au electrodes were grown by thermal evaporation.

A MERLIN Zeiss field emission gun scanning electron microscopy, FEGSEM (Kohen, Germany), instrument was employed for scanning electron microscopy at an accelerating voltage of 5 kV. The monochromator Omni 300 LOT ORIEL, with a single grating in Czerny–Turner optical design, in AC mode with a chopping frequency of 13 Hz and a light bias (1 sun) applied was used for recording the external quantum efficiency. A Keithley 2400 Source Measure Unit and a solar simulator Spectra Physics Oriel 150 W, with an AM1.5G filter set, were used to measure the current–voltage characteristics. Devices were tested inside and outside the glovebox, without encapsulation. Devices were tested with a humidity of 50% under ambient conditions. The reported performances were registered after 30 min in air. Shunt resistance ($R_{sh}$) and series resistance ($R_s$) have been evaluated by assuming the single diode model [44]; $R_{sh}$ can be approximated by the negative of the inverse slope of the characteristic curve where the voltage is null, and $R_s$ can be approximated by the negative of the inverse slope of the characteristic curve where the current is null.

## 3. Results and Discussion

In order to verify the reliability of each set of measurements, devices employing undoped and F4-TCNQ-doped Spiro-OMeTAD were fabricated; LiTFSI-doped Spiro-OMeTAD was also tested as a control device. F4-TCNQ 0.1 wt % doping is effective, compared with undoped Spiro-OMeTAD. The significant increase in the current of the F4-TCNQ-doped films induced an almost two-fold increased efficiency, reaching a similar PCE as the LiTFSI doping method. This result demonstrates the efficient F4-TCNQ doping effect, and notably, the 0.1 wt % device did not present hysteresis (best devices are shown in Table 1). The average PCEs on four devices were 6.38 ± 0.45%, 11.60 ± 0.15%, 8.79 ± 0.30%, 7.63 ± 0.24% and 6.45 ± 0.30% for undoped (0 wt %), 0.1 wt %, 0.5 wt %, 1 wt % and 5 wt % F4-TCNQ doping, respectively. By increasing the molecular doping to 0.5 wt %, a sharp drop in current was clear, which was accompanied by a drop in potential over 1 wt % doping. This decrease in performance was predictable, considering the tendency of F4-TCNQ to create aggregates in mixtures with other molecules [45]. It has been shown that beyond specific doping, the increase in photocurrent is accompanied by an increase in series resistance, caused by insulating aggregates [43,46–48]. Therefore, we can conclude that p-type molecular doping by careful tuning of the dopant weight per cent is promising in enhancing the photovoltaic performance of perovskite cells. Although, for Spiro-OMeTAD, the control LiTFSI-based device exhibited superior performance (short-circuit current density, Jsc = 19.67; open circuit voltage, Voc = 1.08; fill factor, FF = 0.67; PCE = 14.23; hysteresis index HI = 0.00), as already reported previously [38].

**Table 1.** Photovoltaic performance of the best $CH_3NH_3PbI_3$ solar cells varying the p-type doping (where Jsc is the short-circuit current density, Voc is the open circuit voltage, FF is the fill factor, PCE is the photon conversion efficency, and HI is the hysteresis index, defined by Calado et al. [49]).

| HTM | F4-TCNQ Doping (wt %) | $J_{sc}$ (mA cm$^{-2}$) | $V_{oc}$ (V) | FF | PCE (%) | HI |
|---|---|---|---|---|---|---|
| Spiro-OMeTAD | 0.0 | 16.31 | 0.91 | 0.45 | 6.71 | 0.01 |
| | 0.1 | 24.51 | 1.01 | 0.47 | 11.75 | 0.00 |
| | 0.5 | 14.96 | 1.01 | 0.59 | 9.07 | 0.01 |
| | 1.0 | 16.73 | 0.97 | 0.48 | 7.85 | 0.02 |
| | 5.0 | 16.31 | 0.91 | 0.45 | 6.71 | 0.02 |
| P3HT | 0.0 | 16.73 | 0.95 | 0.45 | 7.14 | 0.15 |
| | 0.1 | 15.97 | 0.85 | 0.59 | 8.01 | 0.02 |
| | 0.3 | 16.85 | 0.85 | 0.58 | 8.30 | 0.00 |
| | 0.5 | 19.44 | 0.86 | 0.58 | 9.70 | 0.13 |
| | 0.7 | 16.28 | 0.84 | 0.46 | 6.15 | 0.15 |
| | 1.0 | 14.47 | 0.86 | 0.48 | 5.97 | 0.20 |

We then measured devices with F4-TCNQ:P3HT as the HTM layer. The F4-TCNQ lowest unoccupied molecular orbital (LUMO) is deeper than the P3HT highest occupied molecular orbital (HOMO), inducing efficient electron transfer and an ionised complex (P3HT$^+$) that increases the electrical conductivity [36,45]. The doping effect confirmed in Spiro-OMeTAD was also observed in the P3HT (best devices are shown in Table 1 and Figure 2). The average PCEs on four devices were 7.00 ± 0.19%, 7.90 ± 0.13%, 8.21 ± 0.10%, 9.53 ± 0.24%, 5.80 ± 0.40% and 5.55 ± 0.48%for undoped (0 wt %), 0.1 wt %, 0.3 wt %, 0.5 wt %, 0.7 wt % and 1 wt % F4-TCNQ doping, respectively. The current increased with the dopant concentration up to 0.5 wt % and then it decreased. In the case of 0.1 and 0.3 wt % doping, an excellent balance between current and voltage was achieved, clearly expressed by the improvement of the fill factor (FF). By raising the molecular doping, a worsening in the overall performances occurred. The hysteresis in the current vs voltage measurements reappeared for doping at 0.5 wt %, reaching 0.20 for the 1 wt % doping. For the intermediate doping, 0.3 wt %, the charge extraction has been shown to be so effective that the hysteresis in the measures is squeezed [50–52]. We can assume that, once a certain percentage of doping has been reached, aggregates will start to be created at the interface, affecting the hysteresis. By further increasing the doping, it was evident that the worsening of the performance was due to the short-circuit current lowering and FF reduction (Figure 2).

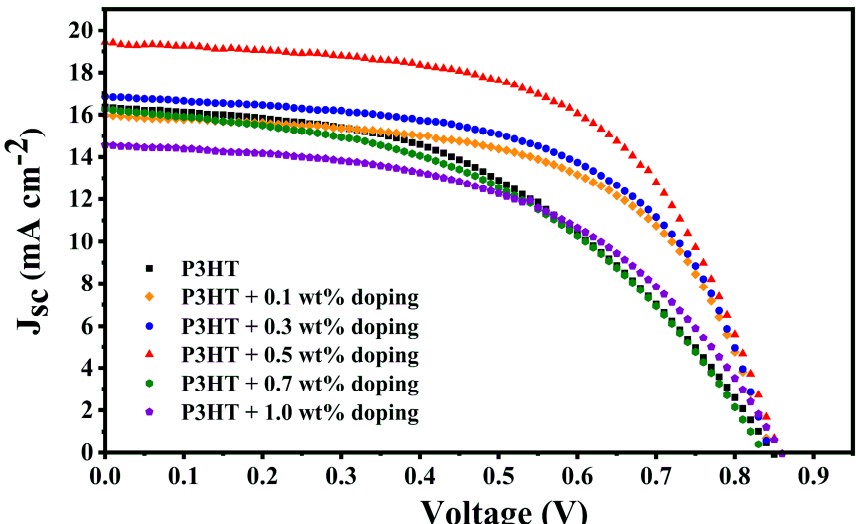

**Figure 2.** Characteristic curves of the best performing devices employing undoped (0 wt %), 0.1 wt %, 0.3 wt %, 0.5 wt %, 0.7 wt % and 1 wt % F4-TCNQ doped P3HT.

The fill factor is affected by the shunt resistance ($R_{sh}$), series resistance ($R_s$), recombination current and reverse saturation current [44]. Here we give a simple evaluation of $R_{sh}$ and $R_s$, taking into consideration the slopes of the characteristic curves where the voltage and current are null, respectively. Even from this trivial evaluation (Figure 3), relevant information can be obtained: (i) $R_{sh}$ remained comparable along the series, except for the worst-performing devices, i.e., 1 wt % doping, in which the shunt resistance decreased, indicating that a leakage path for current flow had arisen [44] and (ii) $R_s$ and HI followed the same trend, supporting the assumption that the optimal doping level is obtained when the increase in the charge extraction is still able to balance the detrimental effect of aggregate formation at the interface between the charge extraction and the active layers.

External quantum efficiency confirmed the photovoltaic performances (Figure 4). The integrated current, $J_{sc}$, was found to be 15.45, 16.57 and 19.18 mA cm$^{-2}$ for undoped P3HT, 0.3 wt % doped P3HT and 0.5 wt % doped P3HT, respectively, ensuring the reliability of the photovoltaic characterisation. The more significant gain that comes from the doping was visible for wavelengths greater than 600 nm; this is compatible with the assumption that the P3HT Fermi level shifts close its HOMO level [53,54].

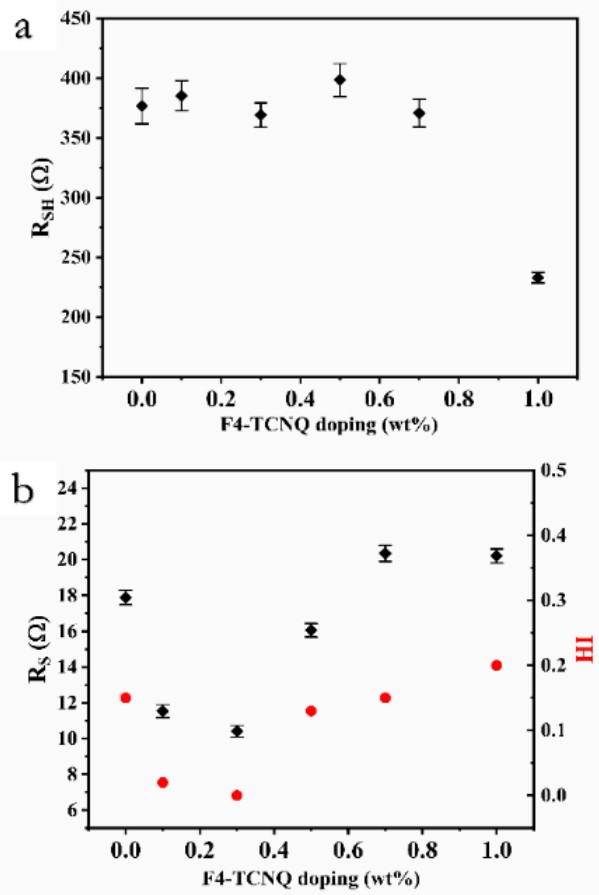

**Figure 3.** (**a**) Shunt resistance, $R_{sh}$, and (**b**) series resistance, $R_s$, evaluated by assuming the single diode model., in devices employing undoped (0 wt %), 0.1 wt %, 0.3 wt %, 0.5 wt %, 0.7 wt % and 1 wt % F4-TCNQ doped P3HT.

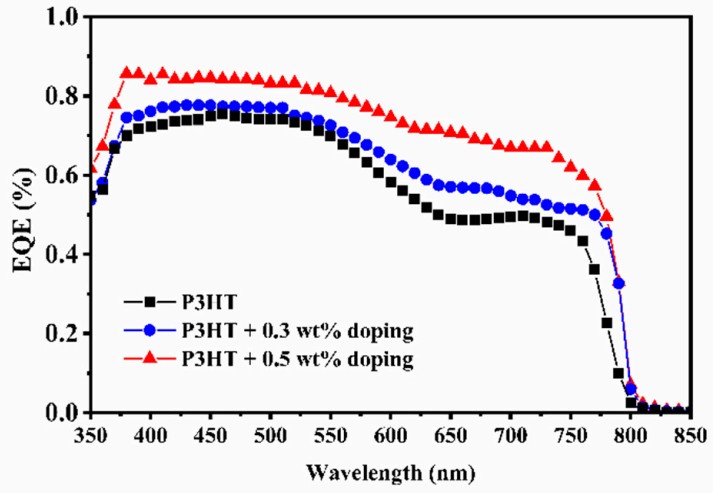

**Figure 4.** External quantum efficiency (EQE) spectra of the best performing devices employing undoped (0 wt %), 0.3 wt % and 0.5 wt % F4-TCNQ doped P3HT.

The characteristic curves' acquisitions in time, in nitrogen and air, and for pristine and 0.3 wt % doped P3HT-based devices are shown in Figure 5. The drastic performance improvement is related to the P3HT interaction with oxygen, quickly forming a charge-transfer complex, that increases the hole concentration [55,56]. The addition of F4-TCNQ led to a further slip of the P3HT Fermi level near

to its HOMO level [53,54], increasing the depletion layer built-in voltage and, therefore, improving the device performances [55]. The doping effect was also apparent when a 0.3 wt % doped P3HT was not exposed to air (Figure 5). The performance was remarkably higher than the pristine one, and the oxygen doping effect on P3HT was mitigated. These results confirm that p-type doping of P3HT with F4-TCNQ acts as an efficient HTM layer in solar-cell perovskites. We suggest that the P3HT stability in air was improved by the P3HT bonding with F4-TCNQ, which started in the precursor solution and evolved in a semi-crystalline phase that may be less inclined to react with oxygen.

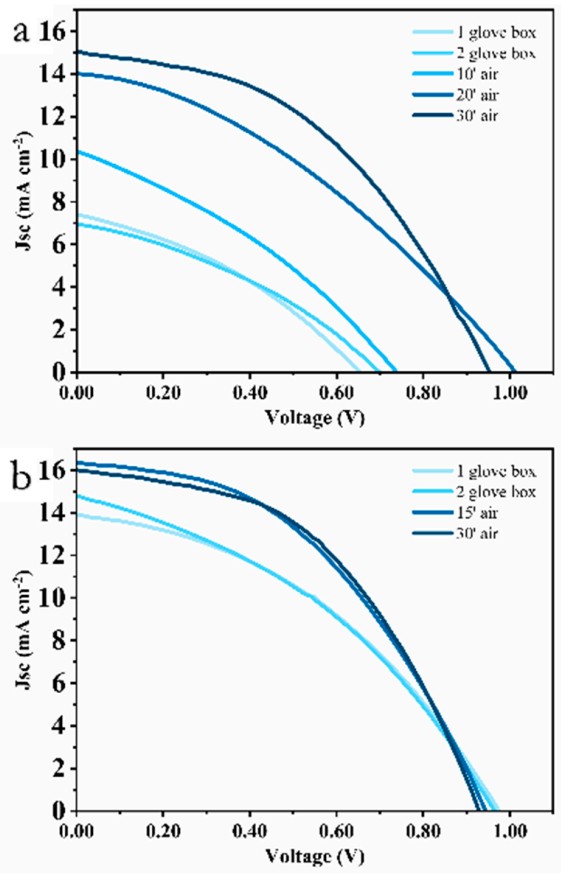

**Figure 5.** Characteristic curves' acquisitions in time for (**a**) pristine and (**b**) 0.3 wt % doped P3HT-based devices.

In Figure 6, the average PCEs in time for four devices employing 0.3 wt % F4-TCNQ:P3HT, stored in the dark in a nitrogen glove box and measured after 30 min in air, are reported. On average they retained 99% PCE after one week and 97% after one month of the first performance, showing remarkable durability.

In order to investigate the nature of this increase in performance, we investigated the optical and morphology properties of the molecular-doped HTM thin film. The UV–Vis absorption spectra (Figure 7) did not show the polaronic bands characteristic of the heavily doped P3HT, which is consistent with the weak doping regime of F4-TCNQ: P3HT films [57]. Note that P3HT–F4TCNQ do form a charged complex even in the weak doping regime [57], which explains the slight improvement in electrical conductivity of F4-TCNQ: P3HT-based devices.

The surface roughness, $R_q$, calculated by analysing the atomic force microscopy, AFM, images in Figure 8, was 11, 6, 8 and 9 nm for the pristine P3HT, 0.3 wt %, 0.5 wt % and 0.7 wt % F4-TCNQ:P3HT, respectively. This slight increase in the film roughness is consistent with previous work on F4-TCNQ:P3HT films in the weak doping regime [57,58]. High dopant concentrations can compromise the P3HT crystalline structure, making it amorphous, and so increasing the scattering centres and the density of traps [53].

Gao et al. [59], reported that the formation of J-aggregate makes the polymer backbone more rigid. They suggested that F4-TCNQ adds holes to the polymer, shifting the equilibrium in favour of aggregated state formation. Considering this and studying the doping aggregation, Jacobs et al. [58] concluded that the segregation of dopant, and the resulting pathways for the charge conduction, is responsible for the higher conductivities but only at low doping levels. In a high doping condition, F4-TCNQ forms isolated domains; therefore, most of the charge carriers remain too strongly bound to be available for the electrical conduction [60]. The reported F4-TCNQ:P3HT-based devices with a dopant concentration higher than 0.5 wt % showed a decrease in performance due to the generation of aggregates acting like traps for the charge carriers, affecting hysteresis and performance. In 0.3 wt % molecular-doped P3HT, the loss in charge carriers caused by the aggregate formation was still well balanced by the charge extraction increase, leading to devices with high performance and without hysteresis.

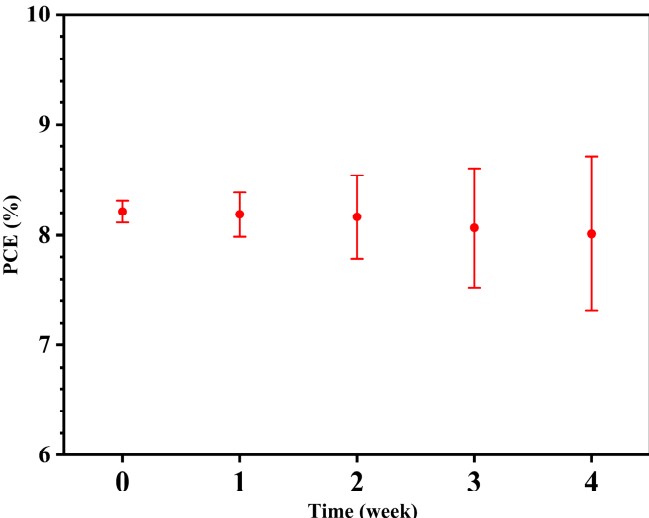

**Figure 6.** Average PCEs in time for four devices employing 0.3 wt % F4-TCNQ:P3HT, stored in the dark in a nitrogen glove box and measured after 30 min in air.

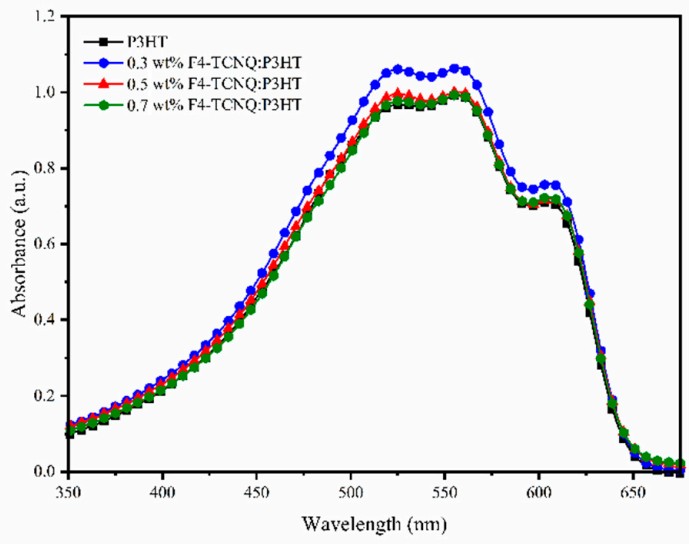

**Figure 7.** UV–Vis absorption spectra of employing undoped (0 wt %), 0.3 wt % and 0.5 wt %, 0.7 wt % F4-TCNQ:P3HT thin film.

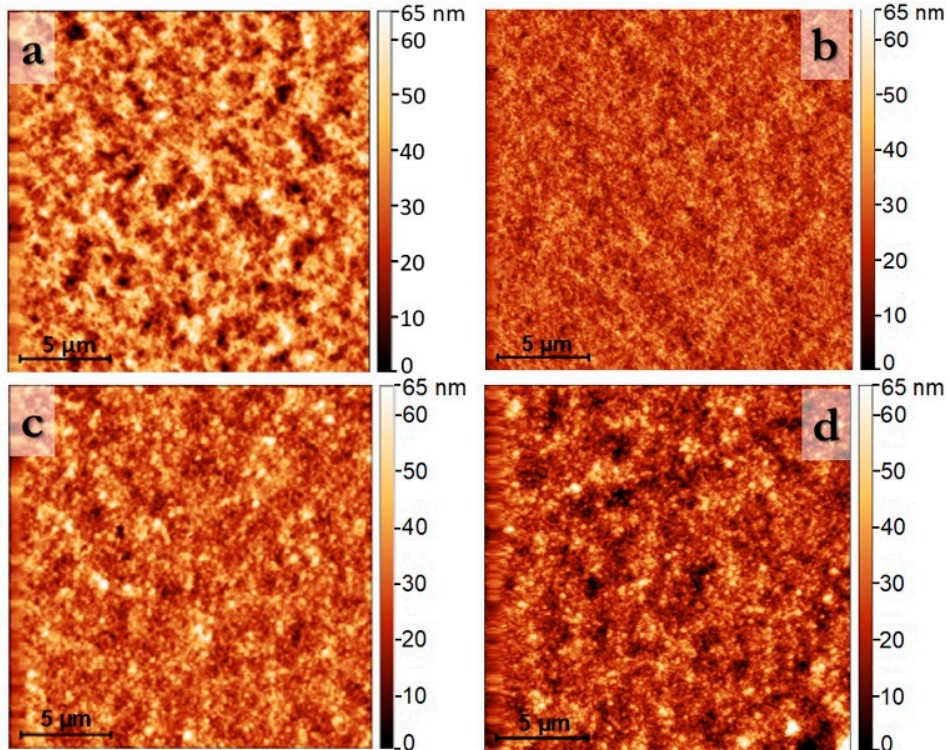

**Figure 8.** Atomic force microscopy, AFM, images of (**a**) pristine P3HT, (**b**) 0.3 wt %, (**c**) 0.5 wt % and (**d**) 0.7 wt % F4-TCNQ: P3HT thin film.

## 4. Conclusions

F4-TCNQ was employed as an additive in a small molecule and polymer HTM. We tested F4-TCNQ doping, in various concentrations, in the c-TiO$_2$: PHJ configuration. We employed a pinhole-free CH$_3$NH$_3$PbI$_3$ layer with reduced surface defects and large grain size, which can generate high photovoltaic performance, not affected by the direction of the voltage scan. We have confirmed that the molecular doping of the HTM is highly effective in using both Spiro-OMeTAD and P3HT but above all that F4-TCNQ employment in P3HT-based devices increases and stabilises performance. F4-TCNQ doping generates the ionised complex P3HT$^+$, which increases the P3HT conductivity and decreases the charge recombination. We have demonstrated that a molecular doping mechanism is more reliable than oxidation doping, using LiTFSI: Spiro-OMeTAD and pristine P3HT. We have shown that F4-TCNQ doping could also mitigate the oxygen effect on P3HT. We attribute the increase in conductivities to the segregation of dopant but only at low doping levels, in order to avoid the formation of aggregates at the interface between the perovskite and the HTM which firstly affects the hysteresis and then the overall performance. We have verified that the employment of F4-TCNQ in P3HT devices stabilises performance, making the solar cell capable of delivering a high performance over a long time. We are confident that fine optimisation and engineering of F4-TCNQ doping in P3HT-based perovskite solar cells would lead to highly efficient devices with long-term stability.

**Author Contributions:** Conceptualisation, V.T., T.D., A.R. and S.C.; methodology, V.T. and S.C.; validation, V.T., A.R. and S.C.; formal analysis, V.T. and T.D.; investigation, V.T. and N.M.; data curation, V.T. and N.M.; writing—original draft preparation, V.T. and T.D.; writing—review and editing, V.T., T.D. and O.F.; visualisation, V.T.; supervision, O.F., A.R. and S.C.; project administration, A.R. and V.T.; funding acquisition, V.T., O.F., A.R. and S.C. All authors have read and agreed to the published version of the manuscript.

**Funding:** V.T. acknowledges the European Union's Horizon 2020 research and innovation programme under the Marie Skłodowska-Curie grant agreement, N° 798271. S.C. acknowledges Regione Puglia and ARTI for funding the future in research (FIR) project "PeroFlex", project no. LSBC6N4. A.R. gratefully acknowledges the Scientific Independence of Young Researchers (SIR) project "Two-Dimensional Colloidal Metal Dichalcogenides-Based Energy-Conversion Photovoltaics" (2D ECO), Bando SIR 2014 MIUR Decreto Direttoriale 23 Gennaio 2014 no. 197

(project number RBSI14FYVD, CUP: B82I15000950008) for funding. O.F. acknowledges the Royal Society for his University Research Fellowship (UF140372).

**Acknowledgments:** The authors acknowledge Sonia Carallo for technical support.

**Conflicts of Interest:** The authors declare no conflict of interest. The funders had no role in the design of the study; in the collection, analyses, or interpretation of data; in the writing of the manuscript; or in the decision to publish the results.

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
