# Peer review of "Molecular Doping for Hole Transporting Materials in Hybrid Perovskite Solar Cells"

_metals, doi:10.3390/met10010014_

Round 1
Reviewer 1 Report
From a submission standpoint, I am a bit unclear why this work was submitted to Metals versus Materials or Inorganics or Nanomaterials. It does not describe a metallic structure, though it contains Pb encased in halides. In this regard, it would have been helpful to see an author's letter describing the relevance of the submission to the journal's focus. This work follows on the author's prior device studies and has a strong device production and function focus. Regardless, it shows a positive response to an molecular fluorine additive that improves long term function and stability of lead halide perovskite photovoltaics without degrading performance. It should be of interest to others working in this competitive field.
1) The introduction needs some improved focus and clarity. It starts off in a very technical manner related to the lead perovskites and organic polymer or molecular additives to improve charge transport, air stability and water repulsion. Then it jumps to an undefined TiO2:PHJ material (what does PHJ mean?)- line 76. It appears that they are talking about TiO2-perovskite hybrids, if so then there should be a clear intro to that subfield before jumping to what they will describe in this work.
2) In Table 1, what does "standard doping mean in column 2 - is there a value? Also should, the first two J entries have periods versus commas (same for PCE entry 1 and Fig. 5 caption)? Are both of the decimal places significant (e.g., 19.67 versus 19.7)? Is Voc for entry 1 really 915 V or 9.15 V?
3) Looking at the Table 1 results, it is difficult to see how the F4-TCNQ has an effect other than lowering PCE in the Spiro system. It appears to have no detrimental effects on photovoltaic properties (small Jsc, Voc, FF changes), thus as a materials protection additive, that may be it key advantage here?
4) In trying to gauge significance of changes in Table 1 or Figure 3, are these average values form several devices or data on single devices? The changes would be more compelling if they represented average/standard deviations for several replicate devices.
5) While there is much electrical and device analysis here, there is not much in the way of microscopic layer analysis. How does F4 interact with the spiro or P3HT in solution (observable UV-vis changes due to charge transfer?), can their interactions be investigated in a pure film? Does F aggregate (EDS/XPS analysis?) or is it uniformly distributed in the dried film?
6) It was unclear why Jsc improves upon air exposure (Fig. 6) what oxidation process occurs and is it reversible?
7) reference 58 has some odd symbols in the title.
Author Response
The authors thank the reviewer for the stimulating report. The manuscript is considerably improved by the useful observations made. Below the answers point by point and references to changes in the main text are listed.
Comments and Suggestions for Authors
Reviewer 1: From a submission standpoint, I am a bit unclear why this work was submitted to Metals versus Materials or Inorganics or Nanomaterials. It does not describe a metallic structure, though it contains Pb encased in halides. In this regard, it would have been helpful to see an author's letter describing the relevance of the submission to the journal's focus. This work follows on the author's prior device studies and has a strong device production and function focus. Regardless, it shows a positive response to an molecular fluorine additive that improves long term function and stability of lead halide perovskite photovoltaics without degrading performance. It should be of interest to others working in this competitive field.
The manuscript has been submitted to the special issue of Metals "Application and Characterisation of Hybrid Halide Perovskites"; the manuscript was prepared to fit most of the topics covered by the collection. (https://www.mdpi.com/journal/metals/special_issues/hybrid_halide_perovskites)
1) Reviewer 1: The introduction needs some improved focus and clarity. It starts off in a very technical manner related to the lead perovskites and organic polymer or molecular additives to improve charge transport, air stability and water repulsion. Then it jumps to an undefined TiO2:PHJ material (what does PHJ mean?)- line 76. It appears that they are talking about TiO2-perovskite hybrids, if so then there should be a clear intro to that subfield before jumping to what they will describe in this work.
The reviewer must be thanked for highlighting this lack: the abbreviation PHJ, planar hetero-junction, was not well clarified. The text has been modified in agreement.
Page 2, lines 76-77:
“in this device architecture, the ETM is only composed by a layer of about 80 nm of titania, and no scaffold is used. (Liu, et al. 2013)”
2) Reviewer 1: In Table 1, what does "standard doping mean in column 2 - is there a value? Also should, the first two J entries have periods versus commas (same for PCE entry 1 and Fig. 5 caption)? Are both of the decimal places significant (e.g., 19.67 versus 19.7)? Is Voc for entry 1 really 915 V or 9.15 V?
The authors are very grateful for pointing out these typos: they have been corrected and the text modified in agreement.
Page 4, lines 154-155
Considering the right observation of the referee, we have removed the values of the doping standard from table 1, and they have been included in the text.
Page 4, lines 150-151 (Jsc = 19,67; Voc = 1.08; FF = 0.67; PCE = 14.23; HI = 0.00)
Moreover, to facilitate reading, the concept of “standard doping” has been replaced throughout the text with “LiTFSI doping”.
Page 2 line 70; page 3 line 89; page 4 lines 137 – 141; page 4 line 151
3) Reviewer 1: Looking at the Table 1 results, it is difficult to see how the F4-TCNQ has an effect other than lowering PCE in the Spiro system. It appears to have no detrimental effects on photovoltaic properties (small Jsc, Voc, FF changes), thus as a materials protection additive, that may be it key advantage here?
The performance improvements are considered compared to the non-doped Spiro system: by removing the performance of the LiTFSI doped Spiro device (the previously called standard doping), table 1 should be more explicit.
4) Reviewer 1: In trying to gauge significance of changes in Table 1 or Figure 3, are these average values form several devices or data on single devices? The changes would be more compelling if they represented average/standard deviations for several replicate devices.
As explained in the main text, the value reported in Table 1 are related to the best devices: we have modified the title of table 1 to underline this.
Page 4, lines 154-155
The authors agree with the referee that average and standard deviations PCE values have been added to the main text
Page 4 lines 143- 144:
The average PCE on 4 devices are 6.38 ± 0.45, 11.60 ± 0.15, 8.79 ± 0.30, 7.63 ± 0.24, 6.45 ± 0.30 for undoped, 0 wt%, 0.1 wt%, 0.5 wt%, 1 wt%, and 5 wt% F4-TCNQ doping respectively.
Page 5 lines 160-162:
The average PCE on 4 devices are 7.00 ± 0.19, 7.90 ± 0.13, 8.21 ± 0.10, 9.53 ± 0.24, 5.80 ± 0.40, 5.55 ± 0.48 for undoped, 0 wt%, 0.1 wt%, 0.3 wt%, 0.5 wt%, 0.7 wt%, and 1 wt% F4-TCNQ doping respectively.
5) Reviewer 1: While there is much electrical and device analysis here, there is not much in the way of microscopic layer analysis. How does F4 interact with the spiro or P3HT in solution (observable UV-vis changes due to charge transfer?), can their interactions be investigated in a pure film? Does F aggregate (EDS/XPS analysis?) or is it uniformly distributed in the dried film?
The report was designed to be part of a special issue about hybrid perovskite applications and about the characterisation methods that can ensure a reliable analysis of device performance. Although the authors find the study path proposed by the referee very interesting and intriguing, they believe that the suggested characterisations should be contained in a separate work.
6) Reviewer 1: It was unclear why Jsc improves upon air exposure (Fig. 6) what oxidation process occurs and is it reversible?
The reversible p-doping of P3HT by oxygen is a well-known phenomenon (e.g. Hintz et al., 2011): the p-doping of the HTM increases the charge extraction mechanism, and so the Jsc improves upon air exposure. The reference has been added to the main text to facilitate reading.
Page 6 line 196; page 13, lines 416-418
7) Reviewer 1: reference 58 has some odd symbols in the title.
Many thanks, the cited reference and others have been corrected.
Page 10 lines 297, 305, 311; page 11 lines 319, 321, 322, 344; page 12 lines 357, 362, 373, 379, 387, 390; page 13 lines 420, 422, 428, 429
Reviewer 2 Report
In this work, the authors demonstrated efficient molecular doping of hole Transport Materials commonly used in perovskite solar cells. To this regard, they employed a strong electron acceptor, namely F4-TCNQ, as an additive in spiro-OMeTAD and P3HT. Optimized dopant concentrations resulted in enhanced cell efficiency and stability. Moreover, they demonstrated that molecular doping is a more effective means to improve solar cell performance compared to oxidation doping occuring in pristine P3HT and in Li-TSFI doped spiro-OMeTAD. The study presented is very thorough and has some novel aspects. The associated discussion and the mechanisms proposed for the improvements seen are generally supported by the data presented and their analyis. Overall, the manuscript is well written and I would rate it as of high quality with some significant results shown, particularly in terms of cell stability over time.
I recommend that the manuscript is published in its current form as it contains no errors or ambiguities and does not require any other data and/or analysis/discussion to be included.
Author Response
The authors thank the reviewer for supporting the work done. We believe that the stability and economy of these solar cells can be their ultimate winning key, and we hope that our report will support research in this direction.
Comments and Suggestions for Authors
Reviewer 2: In this work, the authors demonstrated efficient molecular doping of hole Transport Materials commonly used in perovskite solar cells. To this regard, they employed a strong electron acceptor, namely F4-TCNQ, as an additive in spiro-OMeTAD and P3HT. Optimized dopant concentrations resulted in enhanced cell efficiency and stability. Moreover, they demonstrated that molecular doping is a more effective means to improve solar cell performance compared to oxidation doping occuring in pristine P3HT and in Li-TSFI doped spiro-OMeTAD. The study presented is very thorough and has some novel aspects. The associated discussion and the mechanisms proposed for the improvements seen are generally supported by the data presented and their analyis. Overall, the manuscript is well written and I would rate it as of high quality with some significant results shown, particularly in terms of cell stability over time.
I recommend that the manuscript is published in its current form as it contains no errors or ambiguities and does not require any other data and/or analysis/discussion to be included.
